# Biodegradation of Total Petroleum Hydrocarbons in Soil: Isolation and Characterization of Bacterial Strains from Oil Contaminated Soil

**Runkai Wang [1,2,3], Baichun Wu [1], Jin Zheng [1], Hongkun Chen [1], Pinhua Rao [1,2,3,*], Lili Yan [2,3] and Fei Chai [2]**

1   State Key Laboratory of Petroleum Pollution Control, Beijing 102206, China; wangrunkai@sues.edu.cn (R.W.); wu-baichun@cnpc.com.cn (B.W.); zhengjin2810@163.com (J.Z.); chenhongkun@cnpn.com.cn (H.C.)
2   School of Chemistry and Chemical Engineering, Shanghai University of Engineering Science, Shanghai 201620, China; liliyan@sues.edu.cn (L.Y.); cfpbj0501@163.com (F.C.)
3   Anji Goachieve Environmental Technology Co., Ltd., Huzhou 313300, China
*   Correspondence: raopinhua@sues.edu.cn; Tel.: +86-021-67791211

**Abstract:** In this study, we isolated seven strains (termed *BY1–7*) from polluted soil at an oil station and evaluated their abilities to degrade total petroleum hydrocarbons (TPHs). Following 16 rRNA sequence analysis, the strains were identified as belonging to the genera *Bacillus, Acinetobacter, Sphingobium, Rhodococcus,* and *Pseudomonas,* respectively. Growth characterization studies indicated that the optimal growth conditions for the majority of the strains was at 30 °C, with a pH value of approximately 7. Under these conditions, the strains showed a high TPH removal efficiency (50%) after incubation in beef extract peptone medium for seven days. Additionally, we investigated the effect of different growth media on growth impact factors that could potentially affect the strains' biodegradation rates. Our results suggest a potential application for these strains to facilitate the biodegradation of TPH-contaminated soil.

**Keywords:** total petroleum hydrocarbon; isolation; biodegradation; optimal growth condition; growth media

---

## 1. Introduction

Crude oil, exploited industrially since 1867, has become the world's most important energy resource. Called 'black gold', it is one of the world's most precious commodities, which means that the oil industry plays a vital role in many national economies [1]. In addition, many products and by-products derived from crude oil are widely used: for example, gasoline and diesel oil for transportation; paraffin wax for packaging materials and some cosmetic applications; lubricants, which have many uses including as rust and corrosion inhibitors and engine oils [2].

However, despite the countless advantages of crude oil use, it also is a source of environmental pollution. Total petroleum hydrocarbons (TPHs) are common environmental contaminants originating from crude oil [3]. They include a complex mixture of compounds with carbon numbers ranging from $C_6$ to $C_{35}$, some of which are carcinogenic to humans. TPHs are highly stable, resistant to decomposition, and are highly volatile [4]. As a result, they pose a serious health threat to humans living near the pollution source.

During the last decade, research efforts have focused on the biodegradation of crude oil pollution. Degradation efficiency has been reported to be dependent on the substrate type in addition to many geological, climatological, and ecological factors [5]. Compared with natural attenuation, microbial degradation technology has attracted the most interest, owing to its unique adaptability and

convenience. For example, bacterial strains such as *Rhodococcus rhodochrous*, *Pseudomonas alcaligenes*, *Rhodococcus erythropolis*, and some *Acinetobacter* species have been found to degrade petroleum, depending on the source of carbon and temperature mostly [6–10]. In addition, genera including *Aeromonas, Alcaligenes, Bacillus, Corynebacteria, Flavobacterium, Micrococcus, Mycobacterium, Nocardia, Pseudomonas,* and *Rhodococcus* have been revealed to be capable of degrading aromatic hydrocarbons with a biodegradation efficiency around 60 ~ 70% and biodegradation ability was indicated as an adaptation mechanism. Once the bacteria adapted to the local environment, the metabolism was accelerated and nutrient stimulation was necessary, which could help the microorganism overcome the innate characteristics of nutrient addition to the site, ensuring the sustainability of the long-term process of bioremediation. [11–16].

　　　To build upon this previous work, the aim of this investigation was to discover native bacteria that could be utilized to reduce TPH pollution. In this study, we isolated seven strains from contaminated soil and determined their optimum growth conditions. Furthermore, we identified each of the isolated strains using 16s rRNA sequencing and evaluated their degradation efficiency in different growth media.

## 2. Materials and Methods

### 2.1. Chemicals, Bacterial Isolation and Incubation

　　　Dichloromethane (MC) and sodium sulfate were purchased from SAMCHUN Chemical Corporation. Tryptone, yeast extract, yeast malt, peptone, beef extract and potato dextrose were obtained from Junsei Chemical Co. (Tokyo, Japan). TPH standard solution, NaCl and agar powder were purchased from Sigma-Aldrich (St. Louis, MO, USA). All reagents used were of analytical grade. Bacteria were isolated from Uijeongbu oil station. Supplemented mineral salt (MS) medium was used to isolate the strains for short-term use. This contained the following: $K_2HPO_4$ (0.3 g), $KH_2PO_4$ (0.3 g), KCl (0.1 g), NaCl (0.6 g), $CaCl_2$ (0.1 g), $MgSO_4 \cdot 7H_2O$ (0.1 g), $NH_4Cl$ (0.1 g), trace element solution (1 mL), and TPH standard solution (1 mM) in 1 L of distilled water [17]. A soil sample (5 g) was diluted with 10 mL of distilled water and mixed to homogeneity. All soil samples in the experiment have been sterilized. Then, 1 mL was added into MS medium with 1 mM TPH standard solution and incubated for 3 days at 30 °C. After this time, beef extract peptone (BP) agar medium was used to enrich each strain, which contained the following: 3 g of beef extract, 10 g of peptone, 5 g NaCl, and 20 g of agar in 1 L of distilled water. The cultured solution was sprayed in a thin layer onto the surface of the BP agar medium and incubated at 30 °C for one week. Strains that grew well on the BP agar medium were selected and enriched separately in BP medium. In addition, single bacterial colonies were isolated and cultured at 30 °C in the incubator for longer-term use. All the experiments were repeated three times and the average value was used.

### 2.2. Optimization of Bacterial Growth Conditions

　　　To optimize the growth conditions for each strain in BP medium, the effects of different temperature, pH value, and incubation time on growth conditions were investigated, respectively. Different incubation temperatures (25, 30, 35, and 40 °C) were tested for each strain and colony-forming units (CFUs) were counted following appropriate dilution of the cultures, plating them onto fresh BP agar medium and incubating for 48 h. Similarly, the effects of different pH values (5.0, 6.0, 7.0, 8.0, and 9.0) and incubation times (3, 5, 7, and 9 days) in BP medium were also tested for each strain.

### 2.3. Biodegradation of TPH in the Soil

　　　Growth medium containing the liquid culture (1 mL) was mixed with a soil sample (20 g) in a flask and incubated at room temperature. The concentration of TPH in the soil sample was then analyzed using gas chromatography (GC-2010, Shimadzu, Japan) with a flame ionization detector (FID). The following analytical procedure was utilized: the FID temperature was operated at a temperature

of 320 °C and H$_2$, He, and air flow rates of 40, 40, and 400 mL/min, respectively. A DB-5 column was used and the pressure of the He carrier gas was 59.6 kPa. The total flow rate was 49.8 mL/min and the column flow rate was 1.8 mL/min. The temperature program was 45 °C for 3 min, then the temperature was increased at the rate of 25 °C/min up to 310 °C, with a hold time of 25 min.

All mixtures of soil and bacteria were pretreated prior to GC analysis. First, sodium sulfate was added to remove the moisture completely, followed by MC solution. The mixture was shaken for 2 h and the liquid was then separated using filtration. Next, the solvent in the filtrate containing TPH was removed by evaporation and MC solution was then added to a final volume of 2 mL. The concentration of TPH in each sample was then analyzed by GC.

Additionally, the effect of different growth media on the TPH biodegradative abilities of each isolated strain was investigated using the following media: Luria–Bertani (LB), yeast malt (YM) and potato dextrose (PD). The compositions of the LB and YM media are listed in Table 1. The PD medium was purchased from Junsei Chemical Co. (Tokyo, Japan). Each growth medium was prepared to a final volume of 1 L using deionized water, sterilized at 120 °C for 20 min, and cooled to room temperature prior to use. A single colony of each strain was selected from BP agar and incubated in BP, PD, LB, and YM medium for 7 days at room temperature. Then, 1 mL of each culture was incubated with polluted soil samples and analyzed by GC as described above. Residual TPH concentrations in each soil sample were detected and used to calculate the biodegradation rate for the strains in each experiment.

**Table 1.** Compositions of LB and YM growth media.

|  | Composition | Content (g) |
| --- | --- | --- |
| LB medium | Tryptone | 10.0 |
|  | Yeast extract | 5.0 |
|  | NaCl | 10.0 |
| YM medium | Peptone | 5.0 |
|  | Malt extract | 3.0 |
|  | Dextrose | 10.0 |
|  | Yeast extract | 3.0 |

### 2.4. Identification of the Isolated Strains

Gram stain was performed following a previously tested procedure [18]. For 16S rRNA amplification and sequencing of each strain, the universal primers 27F and 1492R were used for PCR amplification of the 16S rRNA genes [19]. The amplifications were performed using Taq DNA polymerase under standard reaction conditions. Approximately 1500 bp of each amplicon were sequenced using the 27F primer at JEONJU Biomaterials Institute. The sequences were analyzed using the National Center for Biotechnology Information (NCBI) BLAST program. Phylogenetic trees were constructed based on the 16S rRNA gene sequences (500 bp) using the neighbor joining method.

## 3. Results and Discussion

### 3.1. Biodegradation Efficiency of the Isolated Strains in BP Medium

Seven strains, termed *BY1–7*, were isolated from soil from the Uijeongbu oil station using MS and BP medium. As shown in Figure 1A, all strains were able to degrade TPH in MS medium; *BY2* exhibited the highest TPH biodegradation rate (44.7%). In contrast, the TPH concentration naturally decreased by 10.6% over the same time period. These results suggested that the seven isolated strains were capable of biodegradation of the TPH pollutants present in the soil.

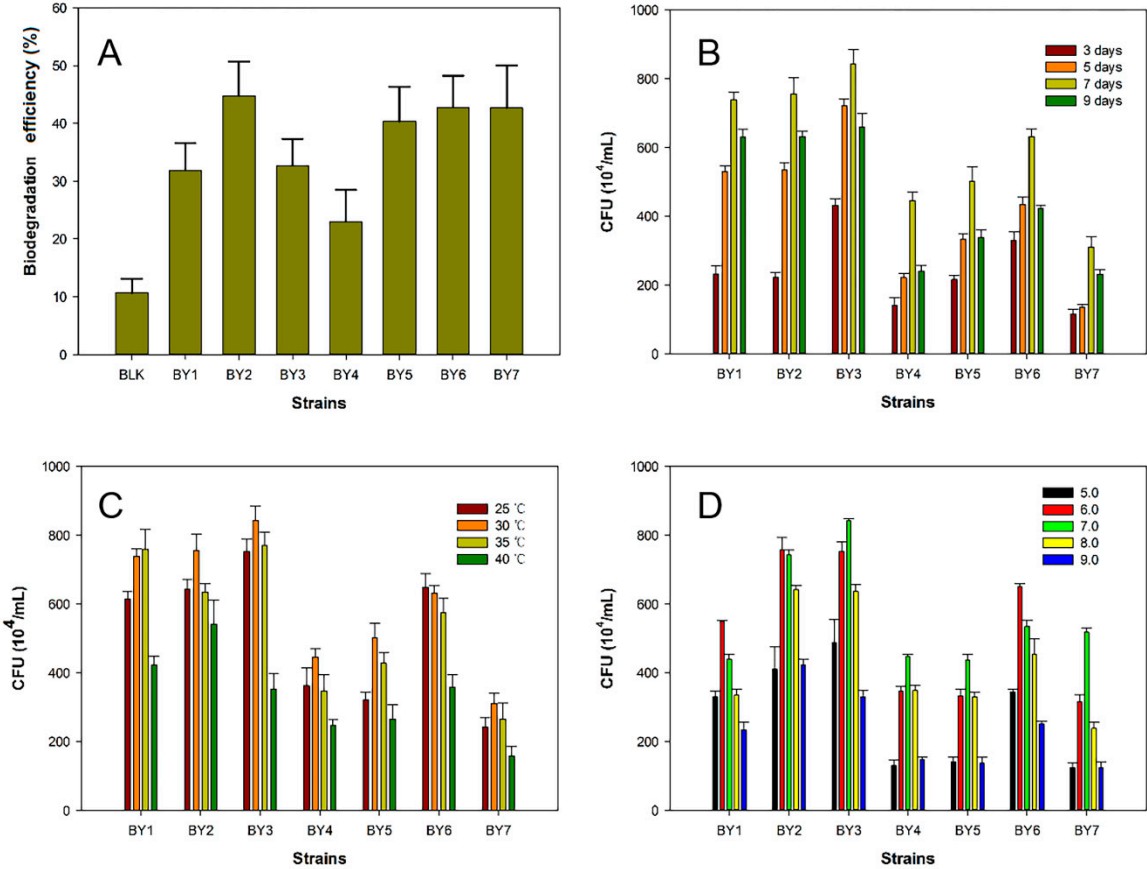

**Figure 1.** Biodegradation efficiency for the isolated strains in MS medium (**A**) and colony numbers under different growth conditions: incubation times (**B**), temperatures (**C**), and pH values (**D**).

### 3.2. Optimization of Growth Conditions of the Isolated Strains in BP Medium

Next, we determined the optimal growth conditions (using various incubation times, temperatures, and pH values) for each of the isolated strains. When different incubation times (3, 5, 7, and 9 d) were investigated, the colony numbers of all seven strains reached their maximum levels after 7 d incubation (Figure 1B). In addition, we examined the effects of different incubation temperatures (25, 30, 35, and 40 °C) in BP medium. A 30 °C temperature was found to yield the most colony forming units for the majority of the strains (Figure 1C), with the exception of *BY6*, which exhibited optimal growth at 25 °C. As shown in Figure 1D, *BY1, BY2,* and *BY6* exhibited optimal growth at pH values around 6, whereas *BY3, BY4, BY5,* and *BY7* yielded the most colony forming units at pH 7. The summarization of biodegradation efficiency, favored temperature and pH of seven strains are listed in Table 2.

**Table 2.** The summarization of biodegradation efficiency of the isolates and its favored temperature and pH respectively.

| Isolates | BY1 | BY2 | BY3 | BY4 | BY5 | BY6 | BY7 |
|---|---|---|---|---|---|---|---|
| Biodegradation Efficiency (%) | 32.1 | 44.7 | 33.2 | 24.5 | 38.2 | 42.3 | 42.1 |
| Most Favored Temperature (°C) | 35 | 30 | 30 | 30 | 30 | 25 | 30 |
| Most Favored pH | 6.0 | 6.0 | 7.0 | 7.0 | 7.0 | 6.0 | 7.0 |

### 3.3. Isolated Strains Identification

Gram–Färbung tests revealed that the majority of the strains were Gram-negative, with the exception of *BY1* and *BY2*, which were Gram-positive (Table 3). Gram-negative bacteria are often harmful to humans' health as many exhibit pathogenic traits.

**Table 3.** Species and Gram–Färbung tests for the isolated strains.

| Strains | Species | Description | Gram–Färbung Test (1) |
|---------|---------|-------------|------------------------|
| BY1 | *Bacillus sp.* | 99.23% similarity to *B. taiwanensis* | + |
| BY2 | *Acinetobacter sp.* | 100.00% similarity to *A. junii* | + |
| BY3 | *Sphingobium sp.* | 100.00% similarity to *S. abikonense* | - |
| BY4 | *Pseudomonas sp.* | 99.86% similarity to *P. mandelii* | - |
| BY5 | *Rhodococcus sp.* | 100.00% similarity to *R. erythropolis* | - |
| BY6 | *Bacillus sp.* | 100.00% similarity to *B. wiedmannii* | - |
| BY7 | *Pseudomonas sp.* | 99.93% similarity to *P. frederiksbergensis* | - |

(1) Symbols denoted the positive (+) and negative (−) in the Gram–Färbung tests.

The isolated strains were identified using 16S rRNA gene sequencing (Figures 2 and 3). *BY1* and *BY6* were found to belong the genus *Bacillus*, while *BY4* and *BY7* were *Pseudomonas* species. Both of these genera have been widely reported to contain bacteria that are capable of degrading compounds including polyvinyl chloride, phthalate ester, eugenol, endosulfan, and dibenzothiophene [20–22]. *BY1* exhibited a 99.2% similarity to *Bacillus taiwanensis*, which is a nitrate-ammonifying bacterium and a partial denitrifier. Previous studies have reported that *Bacillus vireti* is able to degrade benzo[a]pyrene, a common cause of cancer, and lignocellulose [23]. *BY6* exhibited 100.0% similarity to *Bacillus wiedmannii*, which has been reported to be able to degrade polyethylene after 120 days incubation with a 51.53% efficiency and also exhibits biomineralization potential and hydrocarbon-degrading ability [24]. *BY4* was identified as *Pseudomonas mandelii*, which has been reported to contain a hcdABC gene cluster that is necessary for 7-hydroxycoumarin degradation [25]. *BY7* exhibited a 99.9% similarity to *Pseudomonas frederiksbergensis*. Previous studies have revealed that *P. frederiksbergensis* is an effective bioinoculant for enhancing cold stress tolerance in plants and could promote plant growth in soils with high saline levels [26]. However, it has not been previously reported to possess petroleum-degrading ability.

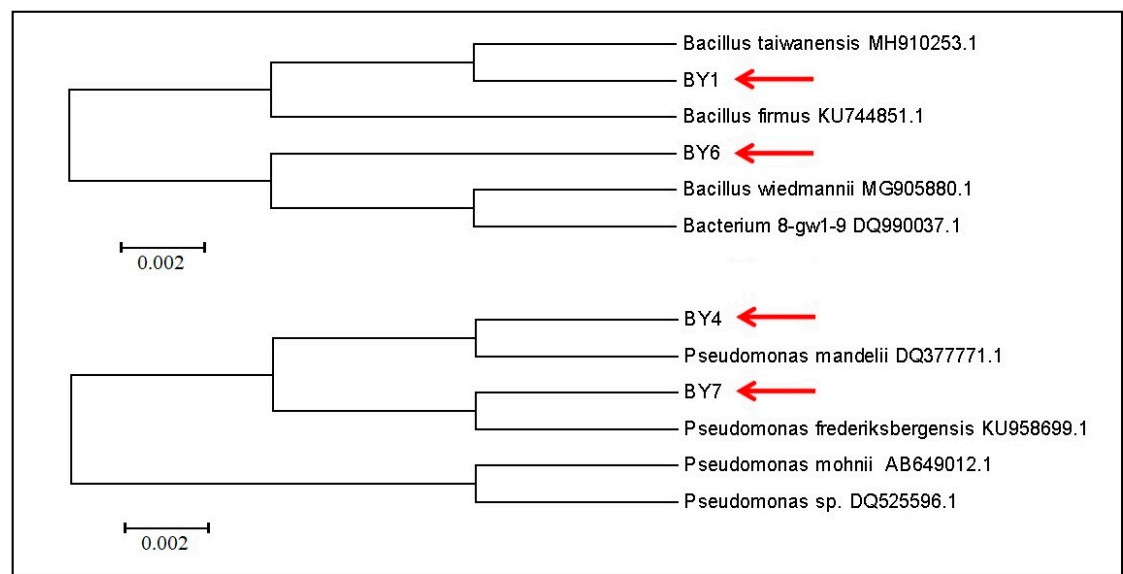

**Figure 2.** Phylogenic tree of the 16S rDNA gene sequence of *BY1, BY4, BY6,* and *BY7* derived by MEGA 4.0 software.

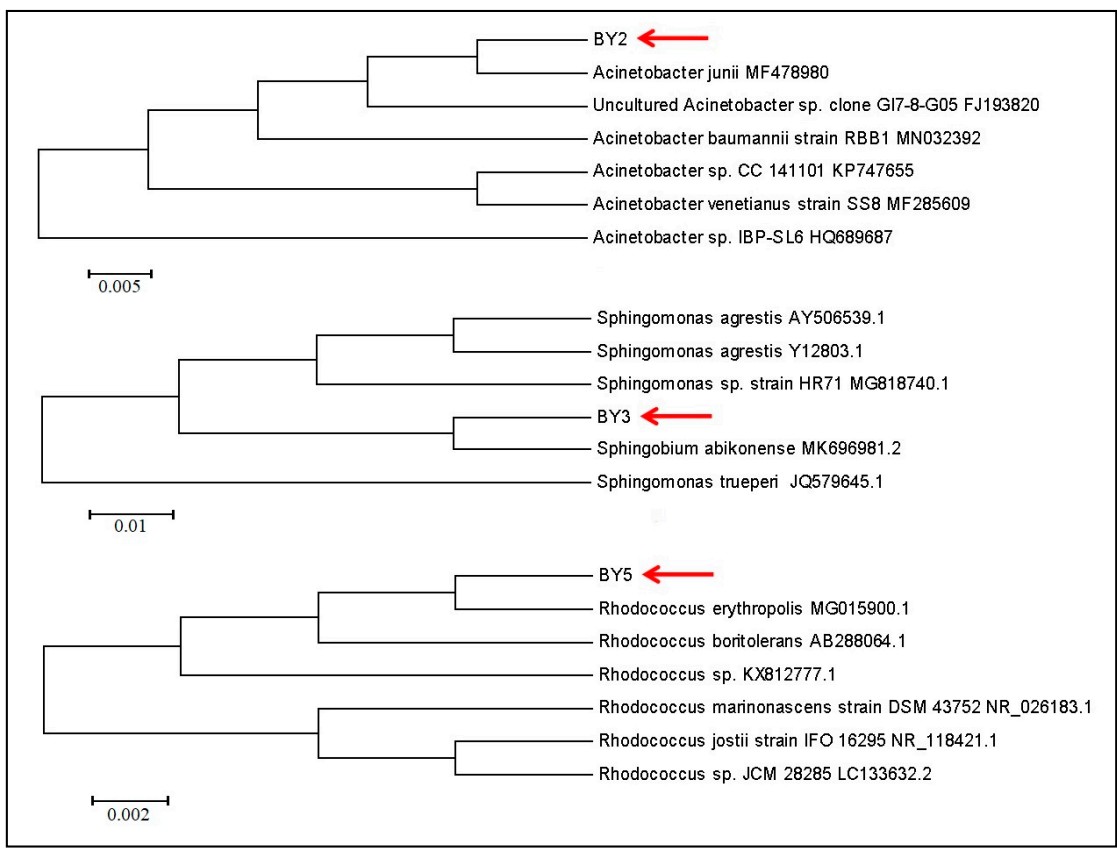

**Figure 3.** Phylogenic tree of the 16S rDNA gene sequence of *BY2, BY3,* and *BY5*.

BY2, *BY3, and BY5* were found to belong to the genera *Acinetobacter, Sphingomonas,* and *Rhodococcus*, respectively. *BY2* exhibited a 100.0% similarity to *Acinetobacter junii*, which is a lead-resistant and surfactant-producing bacterium that is able to degrade crude oil and has a potential application in the bioremediation of oil pollution [27]. *BY3* was revealed to belong to the genus *Sphingobium* and exhibited a 100.0% similarity to *Novosphingobium ginsenosidimutans*, which is a Gram-negative strain that has been reported to be able to grow in a range of different conditions (12–42 °C, pH 5.5–8.5, and 0–1% NaCl) using hydrocarbons as the sole carbon source [28]. *BY5* was found to have a 100.0% similarity to *Rhodococcus erythropolis*, which is an oil-degrading bacterium that produces glycolipid biosurfactants including 2,3,4-succinoyl-octanoyl-decanoyl-2′-decanoyl trehalose and 2,3,4-succinoyl- dioctanoyl-2′-decanoyl trehalose. Interestingly, trehalolipid biosurfactants are produced by *R. erythropolis* at temperatures as low as 10 °C, which suggests a potential role for this strain in bioremediation enhancement in cold regions [9].

### 3.4. Determination of the Biodegradation Efficiency of the Isolated Strains Exposed to Native Soils

The biodegradation efficiency for each of the seven isolated strains was evaluated using the optimal growth conditions that were determined earlier in this study. The biodegradation period in each experiment was varied (0, 3, 5, 7, and 9 days) to identify the time period with the highest biodegradation efficiency. In the uninoculated control experiments, degradation efficiency of background TPH was still decreased due to the effect of native strains present in the original soil. Comparison with blank controls experiment, with the increase of time, the TPH concentration of *BY1–7* gradually decreased, and then, the equilibrium appeared. All strains exhibited an optimal biodegradation efficiency on day 7 (Figure 4A). The residual concentrations of TPH in soil were found to decrease to below 600 mg/kg when incubated with *BY2 (A. junii), BY6 (B. wiedmannii),* and *BY7 (P. frederiksbergensis)*.

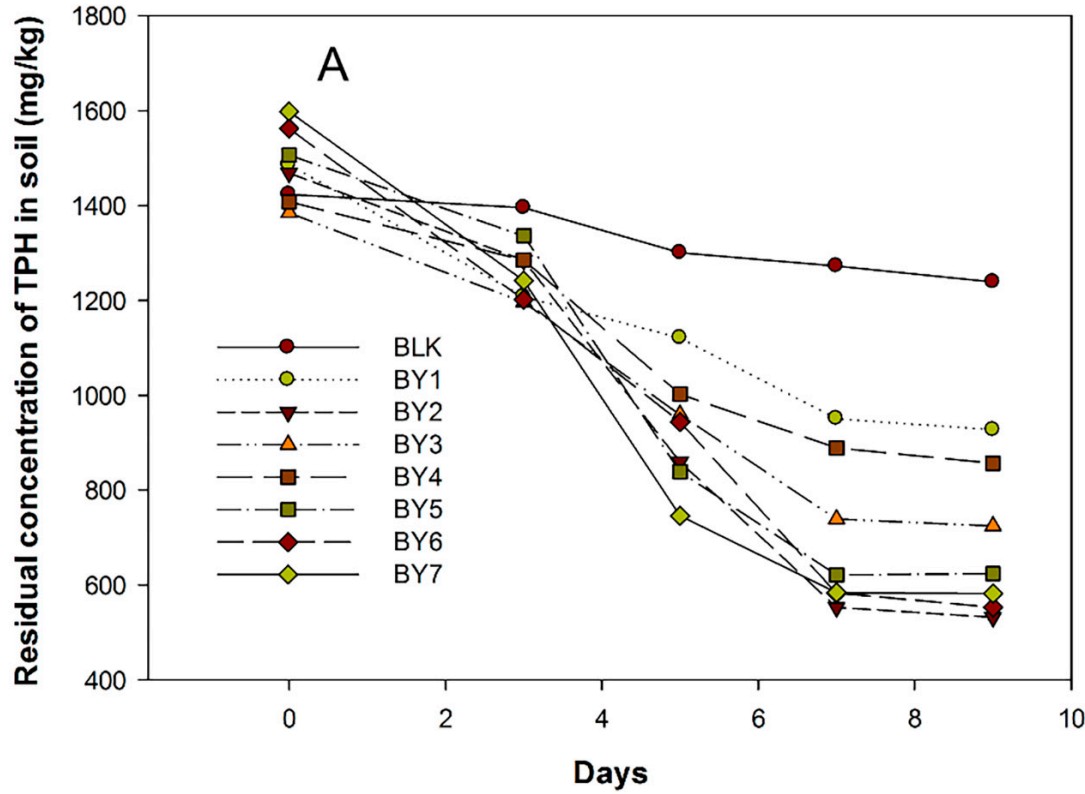

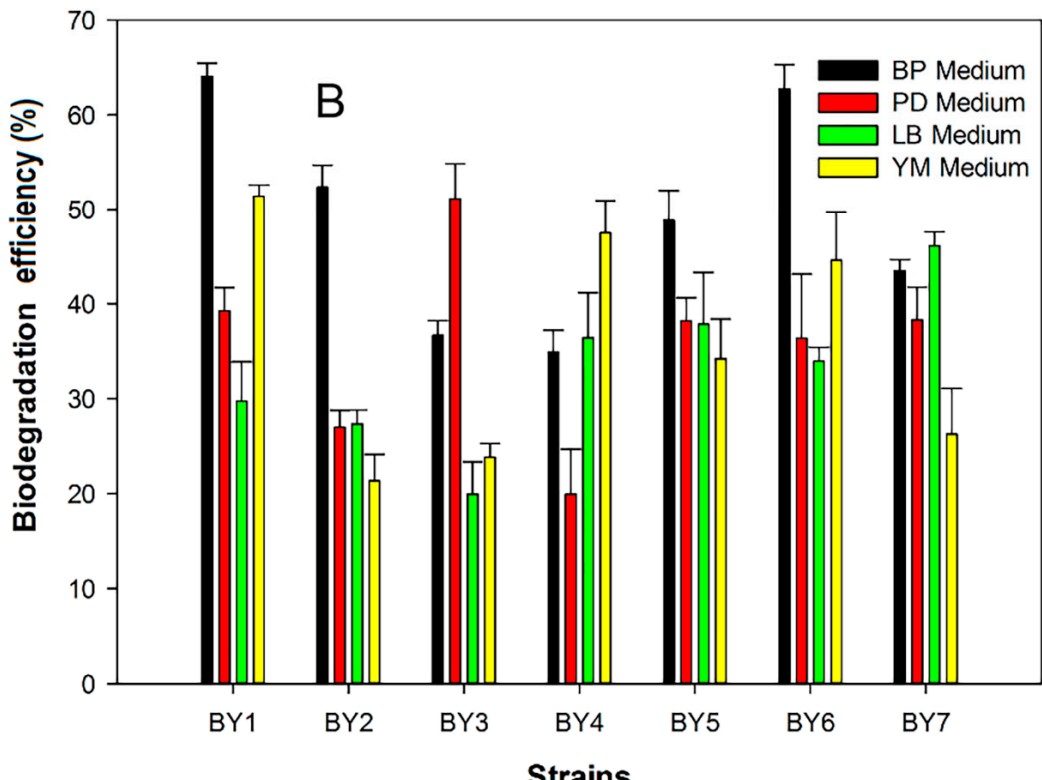

**Figure 4.** Residual concentration of TPH in soils using the isolated strains (**A**) and biodegradation efficiency in different growth media (**B**).

In addition, we examined the effect of different growth media on the degradation efficiency of each strain. As shown in Figure 4B, *BY1 (B. taiwanensis), BY2 (A. junii), BY5 (R. erythropolis),* and *BY6 (B. wiedmannii)* exhibited optimal degradation efficiency in BP medium. *BY3 (S. abikonense)* exhibited its highest degradation efficiency (51.5%). However, *BY4 (P. mandelii)* displayed a maximum biodegradation efficiency of 47.5% and *BY7 (P. frederiksbergensis)* exhibited its highest biodegradation efficiency in LB medium. It is likely that the strains have a better degradation efficiency in some media rather than others, owing to species differences. All of the strains were able to grow in BP medium but it was unable to provide the best conditions for some of them to grow abundantly and exhibit an optimal biodegradation efficiency. Therefore, the selection of an appropriate growth medium is vital to achieve an optimal biodegradation efficiency for any of these strains.

## 4. Conclusions

In this study, we isolated seven bacterial strains from soil around an oil station and found that all exhibited higher "bioaugmentation", which can effectively degrade TPH pollutants. The optimal growth conditions for each of the isolated strains was determined and, after seven days incubation, most strains exhibited an optimal performance level at a growth temperature of 30 °C and a pH value around 7. Our 16S rRNA sequencing identified the strains as belonging to the genera *Bacillus, Acinetobacter, Sphingobium, Rhodococcus,* and *Pseudomonas.* The biodegradation rates of each of the strains were examined in different growth media, which revealed that while many strains preferred the BP medium, *Sphingobium* species exhibited a very high biodegradation efficiency in PD medium. These results highlight a potential application for these strains for the biodegradation of TPH pollutants in soil.

**Author Contributions:** Data curation, R.W. and L.Y.; Formal analysis, B.W.; Investigation, P.R.; Methodology, J.Z. and F.C.; Software, H.C.; Supervision, P.R.; Writing—original draft, R.W.; Writing—review and editing, R.W. All authors have read and agreed to the published version of the manuscript.

**Funding:** This research was funded by the Open Project Program of State Key Laboratory of Petroleum Pollution Control [PPC2018017], CNPC Research Institute of Safety and Environmental Technology.

**Acknowledgments:** This work was technically supported by the Soil Collaborative Innovation Center in Shanghai University of Engineering Science.

**Conflicts of Interest:** The authors declare no conflict of interest.

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
