# Peer review of "Biodegradation of Total Petroleum Hydrocarbons in Soil: Isolation and Characterization of Bacterial Strains from Oil Contaminated Soil"

_applsci, doi:10.3390/app10124173_

Round 1
Reviewer 1 Report
Overall Comments
1) I can see you have properly italicized the genus/species names, however you have missed one on ln220 (Sphingobium) and you have Italicized words next to the genus/species names (eg. Ln 16: respectively is italicized after Pseudomonas). Check this throughout the manuscript. Lastly, there are several occasions where you mention the same genus and use the full name many times. Once you have used the genus name the first time, abbreviation is used after that. (eg. Bacillus taiwanensis, then you can say B. vireti). Check this throughout.
2) You mention each strain was prepared in triplicate. What do you mean? Are you trying to say all experiments were performed in triplicate? If you are, this should be clarified that not just strains were prepared in triplicate. Also, make sure to mention how many replicates you did for each experiment.
Minor Revisions
1) Ln 19: with 50% after incubation. 50% what? I am assuming you mean 50% removal.
2) Ln 27: balck should be black and check the year 1987. I am pretty sure we have been using oil for much longer before that.
3) Lns 48-50: You repeat the same words/phrases here (eg. Once adapted, Once adapted) and the sentence is a little confusing.
4) Ln 67: You told me what the trace element solution was, but you didn’t include it in the paper. Add it or a reference to its composition. Also, you told me about how you isolated the DNA for 16S analysis, but didn’t include it in the manuscript. Please include this information.
5) Ln 68-69: End the sentence after homogeneity. Then All soil samples in the experiment were sterilized.
6) Ln 86: (1 mL) – I’m assuming this means of liquid culture. You should clarify this. Do this throughout the manuscript.
7) Lns 111-115: As I mentioned before, I don’t think you need to describe a Gram stain. This is a standard procedure that can easily be referenced and doesn’t need to take up space in the manuscript.
8) Ln 213-214: “In this study, we isolated seven bacterial strains from soil around an oil station and found that all exhibited has higher " bioaugmentation ", which can effectively degrade TPH pollutants.” The underlined part is confusing. Do you mean all exhibited “high remediation ability”?
Reviewer 2 Report
I read the paper with interest. The authors present their new research about isolation of and characterisation of new bacterial strains from hydrocarbon contaminated soil. The authors used appropriate microbiological techniques for the isolation of the bacteria and used valid molecular techniques such as 16s RNA analysis for further identification and characterisation. Once the bacteria were isolated and characterised, the authors have performed the biodegradation assays using the bacteria exposed to native soil to confirm the capability of the bacteria to breakdown the hydrocarbons. Results are intriguing. The topic is of wide scientific interest. Therefore, based on the academic merit of the work presented, I am comfortable for recommending to accept this work to be published in this journal.
However, I would recommend the authors to review the title, as the title is vague, “Isolation and characterization of the strains from oil contaminated soil”
Authors must say what particular strains are isolated. Perhaps, simply add “bacterial strains”.
Reviewer 3 Report
The article presents very interesting research. Is very good
Author Response
Please see the attachment

This manuscript is a resubmission of an earlier submission. The following is a list of the peer review reports and author responses from that submission.
Round 1
Reviewer 1 Report
This manuscript discusses the identification of hydrocarbon degrading bacteria from soil and their potential use for bioremediation. I think this paper is of interest and relevance as this approach for clean-up of pollutants in gaining interest. I find this paper to be acceptable for publication upon some minor revisions listed below.
Overall Comments
1) Throughout the manuscript, none of the bacterial genus/species names are properly italicized. Make sure to correct all these.
2) The Title: Isolation and Characterization of what?
3) You mention sequences were obtained. Have they been added to GenBank? What are the ascension numbers?
4) Figure 2 and 3: You need to add scale bars for relatedness distance.
5) Throughout the paper, you mention sequence similarities but only full percentages. You need at least the first decimal point (ie. 99.3%) for better understanding of how similar the strains are to their relatives.
Minor Revisions
1) Ln 26: Remove “has been”
2) Ln 27: Replace “Crude oil, or ‘black gold’, is…” with “Also called ‘balck gold’ it is…”
3) Ln 30: Replace “oil for the transportations” with “oil for transportation”
4) Ln 35 and Ln 60-61: C6 to C35 and chemical formulas – the numbers should be subscript
5) Ln 36: Remove the first “are”
6) In Bacterial Isolation and Incubation: What is in the trace element solution? What is your TPH comprised of? If a standard mix, a reference is fine.
7) Ln 79: Remove “1” at start and is your soil sample you mix with sterile?
8) In Identification of the isolated strain: You describe a simple Gram stain in detail, this could just be referenced. Also, how did you isolate the DNA for sequencing? Finally, the section heading should say “strains”
9) Ln 186: Replace “have better” with “have a better”

Reviewer 2 Report
Please see the attached file for my general and specific review comments.
